# Post-myocardial infarction heart failure dysregulates the bone vascular niche

Jedrzej Hoffmann [1,2,3,8], Guillermo Luxán [2,3,4,8], Wesley Tyler Abplanalp[2,3,4,8], Simone-Franziska Glaser [2,3,4], Tina Rasper[4], Ariane Fischer[4], Marion Muhly-Reinholz[4], Michael Potente[5,6,7], Birgit Assmus[1,2], David John[2,3,4], Andreas Michael Zeiher[1,2,3] & Stefanie Dimmeler [2,3,4✉]

The regulation of bone vasculature by chronic diseases, such as heart failure is unknown. Here, we describe the effects of myocardial infarction and post-infarction heart failure on the bone vascular cell composition. We demonstrate an age-independent loss of type H endothelium in heart failure after myocardial infarction in both mice and humans. Using single-cell RNA sequencing, we delineate the transcriptional heterogeneity of human bone marrow endothelium, showing increased expression of inflammatory genes, including *IL1B* and *MYC*, in ischemic heart failure. Endothelial-specific overexpression of MYC was sufficient to induce type H bone endothelial cells, whereas inhibition of NLRP3-dependent IL-1β production partially prevented the post-myocardial infarction loss of type H vasculature in mice. These results provide a rationale for using anti-inflammatory therapies to prevent or reverse the deterioration of bone vascular function in ischemic heart disease.

[1] Department of Cardiology, Center of Internal Medicine, Goethe University Frankfurt, Frankfurt, Germany. [2] German Center for Cardiovascular Research DZHK, Frankfurt am Main, Germany. [3] Cardiopulmonary Institute, Goethe University Frankfurt, Frankfurt, Germany. [4] Institute of Cardiovascular Regeneration, Center of Molecular Medicine, Goethe University Frankfurt, Frankfurt, Germany. [5] Angiogenesis and Metabolism Laboratory, Max Planck Institute for Heart and Lung Research, Bad Nauheim, Germany. [6] Berlin Institute of Health (BIH) and Charité—Universitätsmedizin Berlin, corporate member of Freie Universität Berlin, Humboldt-Universität zu Berlin, Berlin, Germany. [7] Max Delbrück Center for Molecular Medicine (MDC), Berlin, Germany. [8]These authors contributed equally: Jedrzej Hoffmann, Guillermo Luxán, Wesley Tyler Abplanalp. ✉email: Dimmeler@em.uni-frankfurt.de

Bone vasculature provides signals and protection necessary to control stem cell quiescence and renewal[1]. Recent characterization of endothelial cells (EC) in the murine bone led to the identification of at least two functional vessel subsets, based on their differential high (H) and low (L) expression of endomucin (EMCN) and CD31[2–4]. Type H capillaries are arteriole-associated, columnar vessels in the metaphysis and endosteum regions. Type L vessels are sinusoid-associated vessels, which predominate in the whole medullary region[1]. Age-dependent decline of type H endothelium is associated with bone dysregulation and accumulation of long-term hematopoietic stem cells (LT-HSC)[3]. Although the functional interaction between heart and bone is emerging as a trigger of post-infarction inflammation and progression of cardiovascular disease[5], the impact of ischemic myocardial injury and resulting heart failure on the vascular niche in the bone remains unknown.

This study shows that post-MI heart failure drives a loss of type H bone ECs, which is associated with a pronounced inflammatory response and pyroptosis. This change appears to be dependent upon IL-1β and MYC signaling, though independent of B2-adrenergic activity. This investigation provides a rationale for use of anti-inflammatory therapies to prevent the deterioration of the bone vascular niche.

## Results

We induced myocardial infarction (MI) in 12 week-old mice by left anterior descending (LAD) coronary artery ligation[6] (Fig. 1a and Supplementary Fig. 1) and assessed the kinetics of bone EC subsets, hematopoietic progenitor, and stem cells during the development of post-MI heart failure. MI induced a time-dependent reduction of type H EC abundance as assessed by flow cytometric analysis. Type H cells were significantly decreased by day 28 as compared to control mice (Fig. 1b). This observation was confirmed by histological analysis of mouse femurs (Fig. 1d) whereas Type L cells in the diaphysis of the bone were not changed post-MI (Supplementary Fig. 2). The reduction in type H vessels was not age-dependent, as histological assessment confirmed that type H vessel length was significantly shorter 28 days after infarction when compared to age-matched controls (Supplementary Fig. 3b). Coinciding with the decrease of type H endothelium, LT-HSCs significantly increased in mice during the development of post-infarction heart failure (Fig. 1b and Supplementary Fig. 3a). To understand the additional impact on hematopoietic cells in the post-MI bones, we performed a more detailed flow cytometric analysis of the bone hematopoietic stem and progenitor cell compartments (Supplementary Fig. 4a). This analysis revealed a significant expansion of myeloid-biased progenitors, as indicated by the increased CD41 expression at day 28 post-MI (Supplementary Fig. 4b, c). This observation was confirmed by CD41 immunostainings of the femur marrow (Fig. 1d).

To confirm these findings in humans, we determined the number and expression pattern of ECs in bone marrow aspirates of healthy subjects and post-MI heart failure patients (for baseline characteristics see Supplementary Table 1). Flow cytometry analysis showed that elderly heart failure patients have significantly reduced numbers of CD31hiEMCNhi cells—representing type H ECs in the bone marrow aspirates—while the total amount of ECs in these samples was unchanged (Fig. 2a, b). To gain deeper insights into the age-independent regulation of the vascular niche by heart failure, we performed single-cell RNA sequencing of lineage-depleted, CD31+ enriched cells obtained from the bone marrow aspirates of an age-matched healthy volunteer and a patient with post-MI heart failure (Supplementary Table 1). t-stochastic-neighbor-embedding (t-SNE) analysis

revealed 13 clusters with equal distribution of cells from the healthy donor and the heart failure patient among the clusters (Fig. 2c). Cluster annotation (Fig. 2c middle panel) showed that cells expressing EMCN transcripts were significantly enriched in cluster 0, representing the type H EC population (Fig. 2c, d and Supplementary Fig. 5). Re-clustering of these cells revealed a striking divergence in the populations of cells derived from the heart failure patient and the age-matched healthy control (Fig. 2e, left). Analysis of differentially expressed genes between these two new populations did not reveal differences in the expression of endothelial genes like PECAM1 (CD31) (Fig. 2e-right, f). However, inflammatory gene transcripts, specifically IL1B and MYC transcripts, were significantly increased in the heart-failure bone marrow sample (Fig. 2e-right, f). The increase in IL1B and MYC was confirmed at the protein level by immunocytochemistry analysis of bone marrow ECs (Supplementary Fig. 6).

A second in silico approach using pseudotime trajectory analysis of re-clustered EMCN, PECAM1 enriched cells revealed similar findings. We identified six branch points and 13 different states (Fig. 2g), four of which (states 10–13) were predominantly comprised of cells derived from the heart failure patient (Fig. 2g). These states expressed higher levels of IL1B (Fig. 2h). Upregulated genes from state 13 (post-MI heart failure enriched) revealed activation of Wnt and NF-κB signaling pathways (Fig. 2i), which are downstream of IL-1β.

Next, we validated the human data by assessing the impact of MI on the transcriptome of the murine bone marrow vascular niche at the single-cell level (Fig. 3a and Supplementary Fig. 7a, b). EMCN expression was significantly reduced at d28 (Fig. 3b) while the expression of Il1b and Myc are significantly increased post-MI (Fig. 3c). Interestingly, the expression of the IL-1β receptor was particularly enriched in Emcnhigh cells (Supplementary Fig. 7c). The induction of IL-1β protein in type H cells post-MI was confirmed by immunostainings of bone sections (Fig. 3e and Supplementary Fig. 7d). Gene ontology (GO) analysis of the upregulated genes in the murine ECs at d7 post-MI relative to d0 confirmed the inflammatory response (Fig. 3d). Further analysis at d28 relative to d0 revealed suggested increased cell stress and cell death, indicated by enriched GO terms like "Cellular response to stress", "Autophagy" or "Positive regulation of cell death" (Supplementary Fig. 7b). Furthermore, inflammation-related transcripts had a prevailing impact on gene expression signatures (e.g., "Adaptive immune system" or "Leukocyte migration") at d28 (Supplementary Fig. 7b).

The scRNAseq experiments both in humans and in mouse suggested a functional relationship between IL-1β and MYC, which might be implicated in cell death and subsequent loss of type H ECs. Therefore, we used cultured ECs to determine whether IL-1β regulates MYC. Indeed, IL-1β induced an upregulation of the MYC transcript and an increase in the activation of the protein as measured by ELISA in cultured ECs (Fig. 3f). Furthermore, IL-1β treatment-induced pyroptosis in ECs indicated by caspase-1 activation by flow cytometry (Fig. 3g). Since MYC has already been related to cell death[7,8], we tested whether overexpression of MYC is sufficient to induce a loss of type H cell in vivo. To do so, we bred the R26StopFLMYC mouse line that bears a human MYC cDNA preceded by a loxP-flanked "stop" cassette in the Rosa26 locus with Cdh5CreERT2 transgenic animals expressing tamoxifen-inducible Cre recombinase especially in ECs[9] (Fig. 3h). Endothelial-specific overexpression of human MYC in MycEC-OE mutant mice, indeed induced a significant reduction of H-type endothelium (Fig. 3i).

Next, we aimed to define the signals that might link the MI and the bone marrow niche. Two hypotheses were tested: First, we

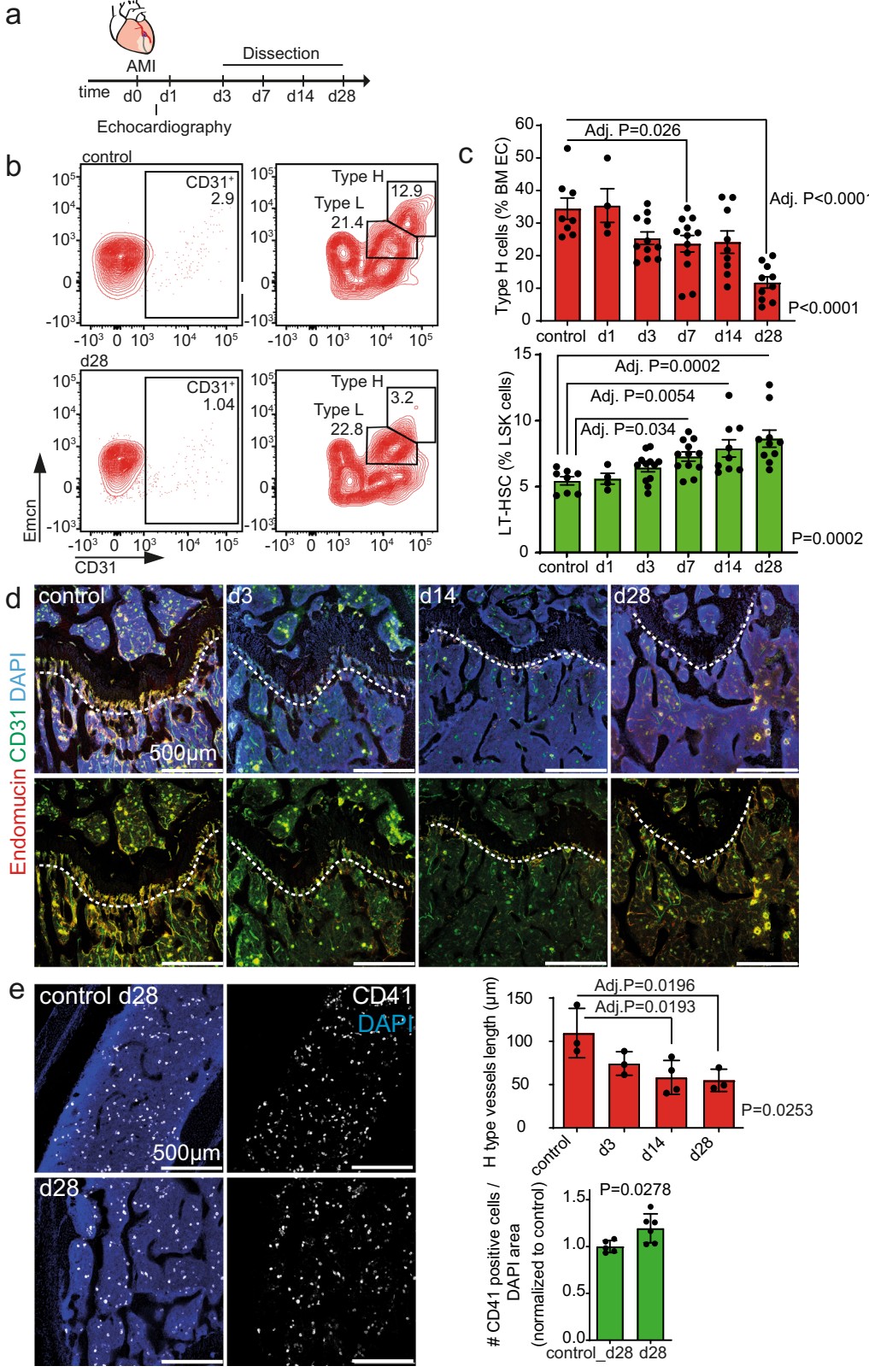

assessed whether acute β-adrenergic stimulation by adrenergic nerves at the level of the bone in the course of MI could be a causative mechanism. A second mechanism might be the systemic inflammatory signaling caused by MI.

Previous studies showed that MI-associated sympathetic activity and stimulation of the bone niche drive an inflammation-mediated deterioration of the niche[10,11]. Moreover, increased β2-adrenergic activity has been shown to substantially influence niche microenvironment by promoting IL-6-dependent CD41+ myeloid progenitor expansion[10]. We, therefore, speculated that blocking beta-2 adrenergic receptor (β2AR) signaling protects the vascular niche in the bone after MI. We performed MI in the

**Fig. 1 Bone type H endothelium is reduced upon myocardial infarction (MI). a** Schematic of the experimental design. **b, c** Flow cytometry analysis of femur bone marrow. Surgery was performed on 12-week-old animals. **b** Gating strategy for endothelial cells (CD45[neg]Ter119[neg] viable single bone cells are shown with further gating on CD31[pos] cells, followed by gating of Type H and Type L EC subsets, based on CD31 and Emcn expression). The numbers shown within the gates represent the percentages of events (cells) within that gates relative to the upstream parent gates (CD45[neg]Ter119[neg] for CD31[pos] cells and CD31[pos] cells for Type H and Type L EC subsets, accordingly). Representative samples from control and day (d) 28 are shown. **c** Quantification. Upper panel, type H endothelial cells are reduced relative to all bone marrow endothelial cells (BM EC) after MI. Lower panel, Long-term hematopoietic stem cells (LT-HSC) are increased relative to LSK cells after MI. $N = 8$ for control, $N = 4$ for d1, $N = 12$ for d3, $N = 12$ for d7, $N = 9$ for d14, and $N = 10$ for d28. Data are shown as mean ± SEM. $P$-value was calculated with ANOVA with Dunnet's multiple comparison test. **d, e** Immunostaining of longitudinal sections through the femur. **d** The length of type H vessels (measured in μm, indicated by the dashed line) is reduced after MI. $N = 3$ for control, d3 and d28, and $N = 4$ for d14. Data are shown as mean ± SEM. $P$-value was calculated with ANOVA. Comparisons among all groups were calculated by Dunnet's multiple comparison test. **e** Myeloid progenitor cell number is increased in the bone marrow 28 days after MI. $N = 5$ for each condition. Data are shown as mean ± SEM. $P$-value was calculated by unpaired, two-tailed Student's $t$-test.

presence of ICI-118,551[12,13], a well-known β2AR antagonist (Supplementary Fig. 8a and Supplementary Fig. 9). However, we could not detect effects on MI-induced reduction of H type endothelium demonstrating that activation of β2AR signaling does not mediate deterioration of the bone vascular niche (Supplementary Fig. 8b, d) or expansion of the myeloid progenitors (Supplementary Fig. 8c, e). This excludes the contribution of sympathetic humoral signaling to the reduction of H-type vasculature after MI.

To test the second hypothesis, we determined whether MI-derived systemic inflammation and induction of IL-1β might play a causal role in mediating the deterioration of the vascular niche in post-MI heart failure. Therefore, we blocked IL-1β production with the selective NLRP3 inflammasome inhibitor MCC950[14] (Fig. 4a). MCC950 treatment partially prevented the MI-induced loss of type H ECs in the bone compared to PBS treated controls as shown by flow cytometry (Fig. 4b) and immunostaining of bone sections (Fig. 4c, d and Supplementary Fig. 10). MCC950 treatment did not have significant effects on the number of LT-HSC (Fig. 4b) but did reduce the number of CD41 positive cells (Fig. 4d).

## Discussion

Here, we show for the first time an impact of post-MI heart failure on the bone marrow vasculature in both mice and humans. The reduction of type H bone ECs seems to be strongly associated with inflammatory responses, as evidenced by the induction of IL-1β in type H vessels, preceding their loss after MI. Pharmacological inhibition of β2-adrenergic activity did not prevent MI-induced deterioration of the vascular bone marrow niche. However, pharmacological inhibition of IL-1β production did protect the bone vascular niche from the detrimental effects of MI, suggesting that inflammatory signaling causes the reduction of H-type vasculature after MI.

Our studies further suggest a crucial role of IL1β-mediated activation of MYC in the H-type vasculature. IL-1β induced MYC expression and is sufficient to cause pyroptosis in ECs in vitro. More interestingly, forced expression of MYC can induce a loss of H-type vessels in the absence of inflammatory signals. Since MYC has been linked to cellular death[7,8], we speculate that inflammatory activation of CD31[hi]EMCN[hi] ECs induces pyroptotic cell death leading to the accelerated depletion of type H endothelium. This might trigger an age-independent disruption of endothelium instructive function in the bone, leading to dysregulation of hematopoiesis and HSC activity, marked by expansion and myeloid lineage skewing of HSCs[15,16].

The NLRP3-inflammasome-dependent pathway might be activated by circulating cytokines or damage-associated molecular patterns (DAMPs) being released from acute and chronically injured cardiac tissue. DAMPs would stimulate NLRP3 activation

in endothelial and immune cells, leading to primary and secondary (by e.g., systemic and paracrine cytokines) IL-1β induction, release, and amplification of inflammatory circuits[17].

Our findings demonstrating that inhibition of IL-1β partially reversed the reduction of type H cells in the vascular niche and inhibited CD41[+] myeloid progenitor cell expansion after MI provides mechanistic support for the therapeutic benefits of the anti-IL-1β antibody canakinumab. Canakinumab administration was shown in a large clinical trial to reduce cardiovascular events in patients with previous MI (CANTOS trial)[18], and more importantly, inhibited the progression of post-MI heart failure[19]. Therefore, pharmacological inhibition of the cytokine IL-1β might be considered as a novel strategy to prevent or reverse the deterioration of the bone vascular function in ischemic heart disease. These findings could also have implications for clonal hematopoiesis of indeterminate potential (CHIP). Since MI can drive myeloid skewing and LT-HSC numbers, it is interesting to speculate whether MI may expedite the expansion of HSCs harboring CHIP-driver mutations. Indeed, in cohorts of patients with ischemic heart failure, it was found that these cohorts have a higher incidence of patients harboring CHIP-driver mutations relative to similar age-matched cohorts published elsewhere[20,21]. Interestingly, an exploratory analysis of patients harboring CHIP mutations (TET2) from the CANTOS trial demonstrated protection from death and hospitalization greater than the non-CHIP population[22].

## Methods

**Mouse strains.** C57Bl/6J mice were used in this study. To generate the endothelial-specific myc overexpressing mice, we bred R26Stop[FL]MYC[9] (MGI ID: 5444670) transgenic animals with Cdh5CreERT2[23] transgenic animals expressing the tamoxifen-inducible Cre recombinase, especially in ECs. The R26Stop[FL]MYC mice bear a human MYC cDNA preceded by a loxP-flanked "stop" cassette in the Rosa26 locus. They are referred to in this work as Myc[EC-OE]. R26Stop[FL]MYC was kept heterozygous for the experimental studies and heterozygous littermates without Cre were used as controls. Cre activity was induced by an injection of tamoxifen (T5648, Sigma) solution every day for 5 consecutive days starting at 8 weeks of age. Following tamoxifen injection of 8-week old mice, hearts were analyzed 4 weeks later. The success of the overexpression was monitored by qPCR by the expression of human Myc with human-specific primers. All animal procedures were performed according to relevant laws and institutional guidelines, were approved by local animal ethics Tierschutzbeauftragte from the Goethe University Frankfurt, and were conducted with permissions FU/1218 and FU/1222 granted by the Regierungspräsidium Darmstadt of Hessen. Mice were held at 23 °C ambient temperature and 60% humidity in 10 h/14 h light/dark cycle.

**Myocardial infarction.** MI was performed in 12-weeks-old male C57Bl/6J mice. Under mechanical ventilation, acute MI was induced by permanent ligation of the LAD coronary artery. MI was confirmed by echocardiography in the first 24 h after the intervention.

**In vivo inhibition of the NLRP3 inflammasome and treatment with β2AR antagonist.** MCC950, a small molecule inhibitor of the NLRP3 inflammasome (S7809; Selleck Chemicals, Houston, USA) diluted in PBS and delivered in vivo at a dose of 5 mg/kg/day via subcutaneous mini-osmotic pumps (1004; ALZET). Mini-

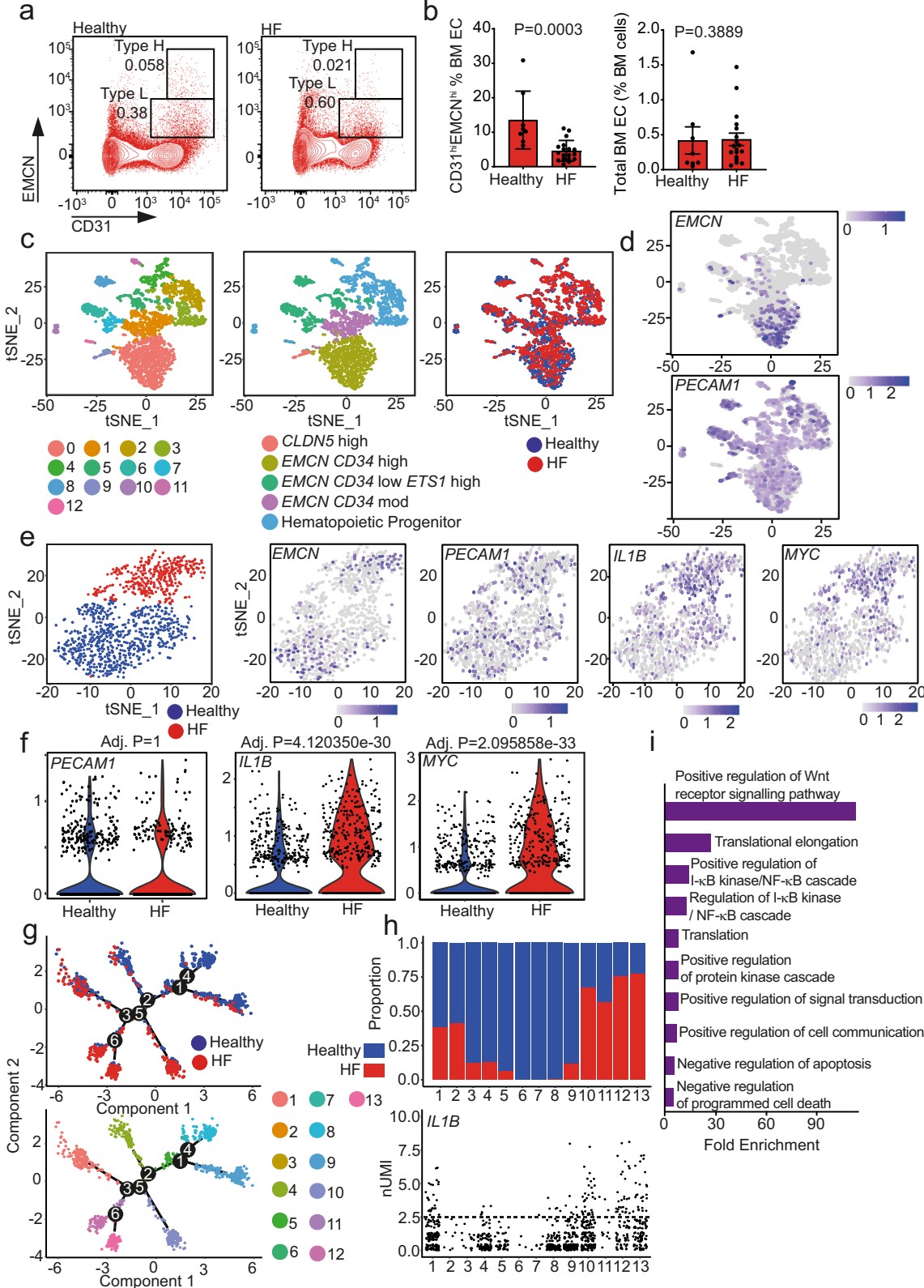

osmotic pumps were implanted in the same intervention where MI was induced. The β2AR antagonist, ICI-118,551 (I127; Sigma Aldrich) was diluted in PBS and delivered in vivo at a dose of 1 mg/kg/day via subcutaneous mini-osmotic pumps (1004; ALZET). Mini-osmotic pumps were implanted in the same intervention where MI was induced. For both interventions, control mice were infused with PBS.

## Flow cytometry of murine samples

*Isolation of mouse bone marrow.* For flow cytometric analysis, tibiae and femurs were collected, cleaned thoroughly to remove the adherent muscles. Cleaned bones were then crushed in ice-cold PBS with mortar and pestle. Whole bone marrow was digested with collagenase II (420U; C2-22; Millipore) at 37 °C for 20 min. After digestion, cells were washed two times with PBS and filtered through a 100 µm cell

**Fig. 2 IL1B is upregulated in *EMCN*-rich endothelial cells in post-myocardial infarction (MI) heart failure (HF). a, b** Flow cytometry analysis of healthy and HF patient bone marrow. **a** Gating strategy for endothelial cells. Representative flow cytometry dot plots showing endothelial cell subsets with a distinct expression of CD31 and Endomucin (EMCN) in healthy and HF bone marrow aspirates (gated on CD45$^{neg}$Lin$^{neg}$ viable single cells). The numbers shown within the gates represent the percentages of events (cells) within that gates relative to the upstream parent gate (CD45$^{neg}$Lin$^{neg}$ cells). **b** Type H endothelial cells are reduced in HF patients compared to healthy controls (left panel), while the total number of endothelial cells remains unchanged (right panel) (N = 8 for healthy, N = 18 for HF patients). BM EC, bone marrow endothelial cells. Data are shown as mean ± SEM. P-value was calculated by unpaired two-tailed, Mann–Whitney test. **c–i** scRNA-seq of a post-MI heart failure patient and age-matched healthy control. **c, d** Clustered cells from both subjects are displayed in t-SNE plots, colored by cluster (left), cell annotation (middle), and health status (right). **d** Expression of *EMCN* and *PECAM1*. *EMCN* is enriched in the cells corresponding to cluster 0. **e–f** Analysis of *EMCN* enriched cell cluster 0 population. **e** Health status-driven dichotomization in *EMCN* enriched population shown in t-SNE plot (Left). Relative expression of key genes in the *EMCN*-enriched population represented by features plots as indicated. **f** Violin plots showing the relative expression of key genes in the *EMCN* enriched population, confirming the significantly increased expression of *IL1B* and *MYC* in the HF sample. P-value was calculated by a two-tailed non-parametric Wilcoxon test with Bonferroni correction. **g** Distribution of cells along pseudotime trajectory branchpoint. Pseudotime analysis revealed 13 states. **h** Distribution of cells among pseudotime states and relative *IL1B* expression. Distribution analysis revealed that states 10, 11, 12, and 13 consist mainly of HF patient cells. *IL1B* expression is higher in states 12 and 13. The dashed line indicates normalized Unique Molecular Identifier (nUMI) counts of 2.5. **i** Gene ontology term ranking of upregulated genes in pseudotime state 13.

strainer (EASYstrainer™ 100 μm; Greiner Bio-One). The cell concentration was adjusted to 10$^6$/100 μl in cell stain buffer (Biolegend).

*HSC panel.* HSC lineages were analyzed similarly to Hérault et al.[24]. Equal number of cells were stained with FITC-conjugated Lineage cocktail (133301; Biolegend; clones 145-2C11, RB6-8C5, M1/70, RA3-6B2, Ter-119) (1:10), BV421- conjugated c-kit (105827; Biolegend; clone 2B8) (1:20), PE-Cy7- conjugated Ly-6A (122513; Biolegend; clone E13-161.7) (1:50), APC-conjugated CD48 (103411; Biolegend; clone HM48-1) (1:50), PE-conjugated CD150 (115903; Biolegend; clone TC15-12F12.2) (1:50), BV-510 conjugated CD41 (133923; Biolegend; clone MWReg30) (1:50) and APC-Cy7-conjugated CD16/32 (101328; Biolegend; clone 93) (1:100) for 30 min on ice. After washing and adding 7AAD viability staining solution (420404; Biolegend) (1:20), cells were acquired on BD FACS Canto II flow cytometer and analyzed using FlowJo (Version 10; FlowJo LLC).

*EC panel.* The equal number of cells were then blocked with 2 μl FC-Block (Mouse BD Fc Block™) and 2.5 μl rat serum (10710 C; Thermo fisher scientific) per 100 μl cell suspension for 10 min. After blocking, cells were stained with FITC-conjugated Ter119 (116206; Biolegend) (1:20), rabbit anti Ephrin-B2 (ab131536; abcam) (1:50), eFluor 660-conjugated anti EMCN (50-5851-82; Thermo fisher scientific; clone V.7C7) (1:20), BV510-conjugated CD45 (103137; Biolegend; clone 30-F11) (1:20) and PE-Cy7-conjugated CD31 (102418; Biolegend; clone 390) (1:50) for 30 min on ice. After washing two times with cell stain buffer, secondary antibodies (BV421-conjugated donkey anti-rabbit (406410; Biolegend)) (1:50) were added in brilliant stain buffer (BD) and 2.5 μl donkey serum (017-000-121; Dianova) and incubated for 30 min on ice. After a further washing step with cell stain buffer (420201; BioLegend), 7AAD viability staining solution (420404; Biolegend) (1:20) was added, and cells were acquired on BD FACS Canto II flow cytometer and analyzed using FlowJo (Version 10; FlowJo LLC).

**Heart failure patients.** Bone marrow aspirates were obtained from healthy volunteers without any evidence of coronary artery disease in their history (N = 8, median age 31 years [IQR 27–37]) as well as from patients suffering from post-infarct chronic heart failure (CHF) with severely reduced left ventricular function, undergoing intracoronary infusion of bone marrow mononuclear cells within the REPEAT trial (Repetitive Progenitor Cell Therapy in Advanced Chronic Heart Failure; NCT 01693042, N = 19). All controls and patients provided written informed consent. The ethics review board of the Goethe University in Frankfurt, Germany, approved the protocols (Approval no. 160/15 for healthy controls and the study consent for REPEAT trial)[25]. The study complies with the Declaration of Helsinki. Patients were eligible for inclusion into the study if they had stable CHF symptoms described by New York Heart Association (NYHA) classification of at least II, had a previous successfully revascularized MI at least 3 months before bone marrow aspiration, and had a well-demarcated region of left ventricular dysfunction on echocardiography. Exclusion criteria were the presence of acutely decompensated heart failure with NYHA class IV, an acute ischemic event within 3 months prior to inclusion, a history of severe chronic diseases, documented cancer within the preceding 5 years, or unwillingness to participate.

**Flow cytometry of human bone marrow.** In total, 100 μl of bone marrow was blocked with 2 μl Fc Receptor Blocking Solution (Human TruStain FcX™; 422301; Biolegend) for 10 min at room temperature. BV-421-conjugated CD31 (303124; Biolegend; clone WM59) (1:20), APC-Cy7-conjugated CD45 (368516; Biolegend; clone 2D1) (1:20), FITC-conjugated Lineage cocktail 4 (562722; BD; clones RPA-

2.10, HIT3a, RPA-T4, M-T701, HIT8a, B159, GA-R2 (HIR2)) (1:5), APC-conjugated AC133 (130-090-826; Miltenyi Biotec) (1:10), PE-Cy7-conjugated CD34 (343516; Biolegend; clone 581) (1:20) and biotin-conjugated EMCN (ab45772; abcam; clone TX18) (1:10) were added to the bone marrow and incubated for 20 min. After staining BM was washed two times with cell stain buffer followed by a second incubation step with PE-conjugated Streptavidin (405204; Biolegend) (1:50) for 20 min. Erythrocytes were lysed with 1× RBC lysis buffer (420301; Biolegend) for 10 min and washed two times with cell stain buffer. After adding 7AAD viability staining solution (420404; Biolegend) (1:20), cells were measured on BD FACS Canto II flow cytometer and analyzed using FlowJo (Version 10; FlowJo LLC).

**Enrichment of human bone marrow ECs for RNA sequencing.** Lineage negative and CD31 positive BMCs were isolated using the first immunomagnetic Lineage Cell Depletion Kit (130-092-211; Miltenyi Biotec) followed by positive selection with CD31 MicroBead Kit (130-091-935; Miltenyi Biotec) using a magnetic cell separation device (QuadroMACS Separator;130-090-976; Miltenyi Biotec). Briefly, BMC suspensions were incubated for 10 min at 4 °C with Biotin-Antibody Cocktail, washed with MACS Buffer, and then incubated for 15 min with anti-Biotin MicroBeads. After washing, cells were separated on LS columns (130-042-401; Miltenyi Biotec) for positive and negative selection, respectively, according to the manufacturer's instructions. After counting lineage negative cells, a second separating step with CD31 followed. Lineage negative cells were incubated with FcR Blocking Reagent and CD31 MicroBeads for 15 min at 4 °C. After washing, cells were separated on LS columns (130-042-401; Miltenyi Biotec) for positive and negative selection, respectively, according to the manufacturer's instructions. Purity (Lineage and CD31 expression) of sorted fractions was checked by FACS-analysis with BV421-conjugated CD31 (303124; Biolegend; clone WM59) antibody and FITC-conjugated Lineage cocktail 4 (562722; BD; clones RPA-2.10, HIT3a, RPA-T4, M-T701, HIT8a, B159, GA-R2 (HIR2)).

**Human bone marrow immunocytochemistry.** Human bone marrow ECs were MACS-enriched from freshly drawn BM aspirates (in parallel from healthy control and from a CHF patient). The cell suspension was adjusted to 50,000 cells/100 μl. 200 μl of cell suspension (100.000 cells) were transferred into a Cytofunnel chamber and centrifuged for 5 min at 500×g, to allow complete fluid absorption. Air-dried samples were fixed with 4% PFA, washed with PBS, and permeabilized with 0.1% Triton X in PBS for 15 min at room temperature. After blocking, samples were incubated with primary antibody overnight at 4 °C, followed by three wash steps (0.05% Triton X in PBS) and incubation with the secondary antibody for 1 h at room temperature.

*Primary antibodies.* Mouse anti-Myc (MA1-980; Thermofisher) and Biotin anti-IL-1β (511703; Biolegend).

*Secondary antibodies.* Goat anti-mouse Alexa Fluor 555 (A-21425; Life Technologies) and Streptavidin Alexa Fluor 555 (S21381; Life Technologies).

**Immunostaining.** Freshly dissected bone tissues were fixed in ice-cold 4% paraformaldehyde solution for 4 h. After PBS washing, decalcification was carried out with 10% EDTA/Tris/HCL pH = 7.0 (15575-038; Invitrogen) at 4 °C with constant shaking for 48 h and after three times washing with PBS, decalcified bones were immersed into 20% sucrose (S0389; Sigma) and 1% polyvinylpyrrolidone (PVP) (P5288; Sigma) solution for 24 h. Finally, the tissues were embedded and frozen in

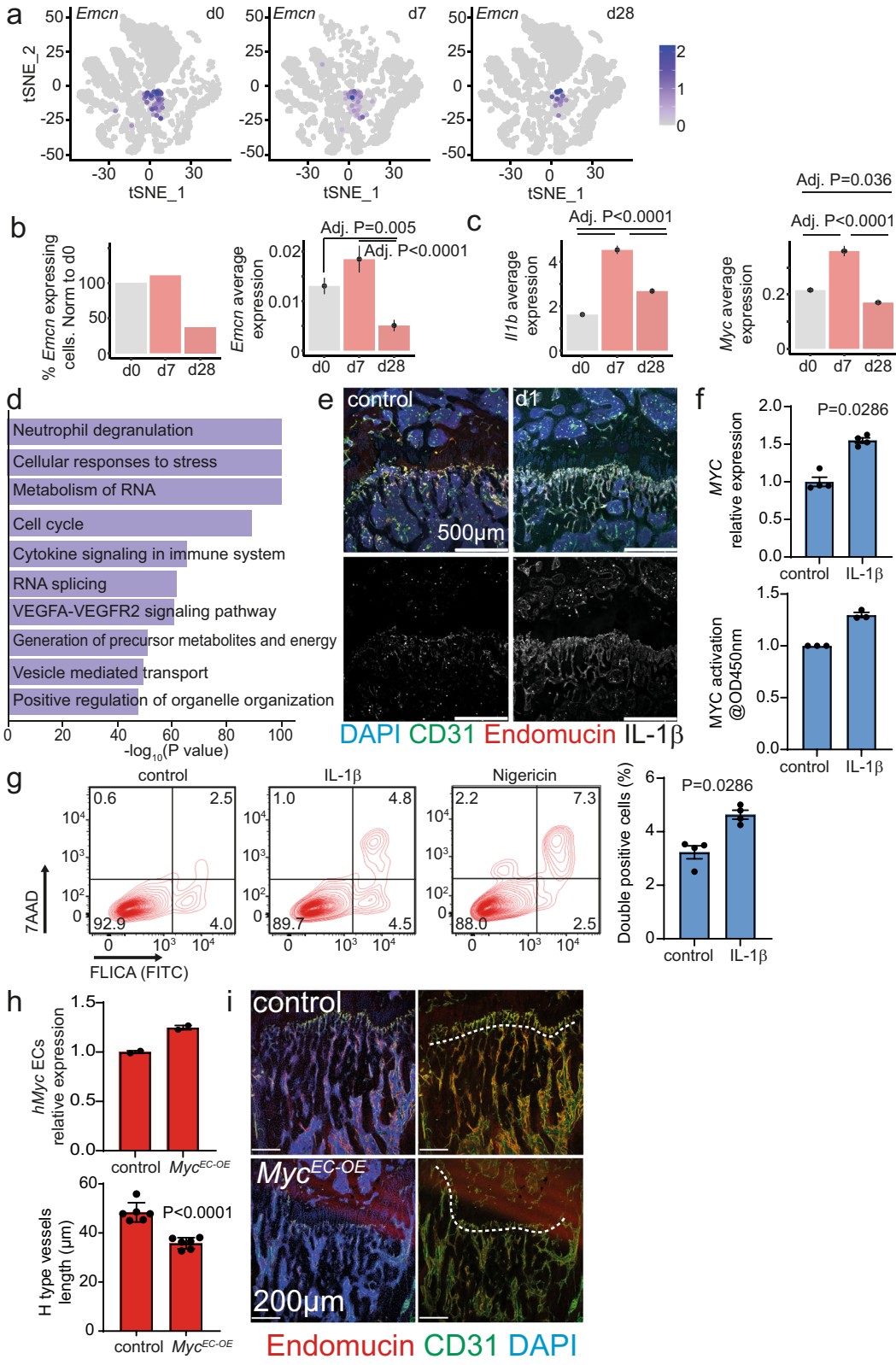

8% gelatin (porcine) (G1890; Sigma) in presence of 15% sucrose and 1% PVP. Cryosections were produced at a Leica CM3050S cryostat

Cryosection was stored at −20 °C. Before starting the staining procedure, they were allowed to acclimate to room temperature. Slides were then rehydrated in PBS and permeabilized with PBS containing 0.3% Triton X-100 by washing them three

times for 10 min. Tissue was blocked for one hour in freshly prepared blocking solution (3% BSA, 0.1% Triton X-100, 20 mM MgCl₂, and 5% donkey serum in PBS). Primary antibodies were incubated overnight at 4 °C in a blocking solution. On a consecutive day, the primary antibodies were washed four times 5 min in PBS. Secondary antibodies were incubated for one hour at room temperature in PBS

**Fig. 3 IL-1β induces MYC and pyroptosis in endothelial cells. a** scRNA-seq of mice bone marrow vascular niche 7 (d7) and 28 days (d28) after myocardial infarction as well as of controls (d0). Clustered cells from the three-time points are displayed in t-SNE plots. *Emcn* expressing cells are colored. **b**-left Representation of the percentage of *Emcn* expressing cells of the lineage depleted, CD31 expressing population. **b**-right *Emcn* and (**c**) *Il1b* and *Myc* average expression. **d** Representation of the ten most significant upregulated terms at d7 revealed by gene ontology analysis when comparing d0 to d7. **e** Immunostaining of longitudinal femur sections at d1. IL-1β immuno-signal is enriched in type H vessels one day after the ischemic insult. **f**-up RT-qPCR analysis of *MYC* expression in HUVEC cells after 1 h IL-1β treatment. $N = 4$. Data are shown as mean ± SEM. *P*-value was calculated by a two-tailed Mann–Whitney test. **f**-down Analysis of MYC activation in HUVEC cells after 1 h IL-1β treatment. $N = 3$ independent experiments with two technical replicates. Data are shown as fold-change relative to control. **g** Analysis of Caspase1 activation in HUVECs after 1 h IL-1β treatment or nigericin as a positive control. Left, Gating strategy. Right, quantification. $N = 4$. Data are shown as mean ± SEM. *P*-value was calculated by a two-tailed Mann–Whitney test. **h** Analysis of human *MYC* expression in isolated liver endothelial cells by RT-qPCR. $N = 2$ for both groups. Data are shown as mean ± SEM. **i** Immunostaining of longitudinal sections through the femur. The length of type H vessels (indicated by the dashed line, endomucin in red, CD31 in green, and DAPI in blue) is reduced in $Myc^{EC-OE}$ mice when compared to controls. Tamoxifen was given at 8 weeks and analysis performed 4 weeks later, $N = 6$ for both groups. Data are shown as mean ± SEM. *P*-value was calculated by unpaired, two-tailed Student's *t*-test.

containing 5% BSA. Immunostainings were imaged in a Leica SP8 confocal inverted microscope.

*Primary antibodies*. Goat anti CD31 Alexa Fluor 488 conjugated (FAB3628G; R&D) (1:50), rat anti EMCN (sc-65495; Santa Cruz) (1:100), rat anti CD41 Alexa Fluor 647 conjugated (133934, Biolegend) (1:100), and mouse anti IL-1β (12242; Cell Signaling) (1:100).

*Secondary antibodies*. Donkey anti rat Alexa Fluor 594 (A21209; Life Technologies) (1:100) and donkey anti mouse Alexa Fluor 647 (A31571; Life Technologies) (1:100).

Stainings were quantified using Volocity Software 6.5.1 (Quorum Technologies).

**Single-cell RNA sequencing library preparation**. Cellular suspensions were loaded on a 10X Chromium Controller (10X Genomics) according to the manufacturer's protocol based on the 10X Genomics proprietary technology. Single-cell RNA-Seq libraries were prepared using Chromium Single Cell 3′ Reagent Kit, v2 (human) and v3.1 Next GEM (murine) (10X Genomics) according to manufacturer's protocol. Briefly, the initial step consisted of performing an emulsion capture where individual cells were isolated into droplets together with gel beads coated with unique primers bearing 10X cell barcodes, UMI (unique molecular identifiers (UMI)), and poly(dT) sequences. Reverse transcription reactions were engaged to generate barcoded full-length cDNA followed by the disruption of emulsions using the recovery agent and cDNA clean up with DynaBeads MyOne Silane Beads (Thermo Fisher Scientific). Bulk cDNA was amplified using a Bio-metra Thermocycler Professional Basic Gradient with 96-Well Sample Block (98 °C for 3 min; cycled 14×: 98 °C for 15 s, 67 °C for 20 s, and 72 °C for 1 min; 72 °C for 1 min; held at 4 °C). Amplified cDNA product was cleaned with the SPRIselect Reagent Kit (Beckman Coulter). Indexed sequencing libraries were constructed using the reagents from the Chromium Single Cell 3′ v2 or v3.1 Next GEM Reagent Kit, as follows: fragmentation, end-repair, and A-tailing; size selection with SPRIselect; adaptor ligation; post-ligation cleanup with SPRIselect; sample index PCR and cleanup with SPRI select beads. Library quantification and quality assessment were performed using Bioanalyzer Agilent 2100 using a High Sensitivity DNA chip (Agilent Genomics). Indexed libraries were equimolarly pooled and sequenced on two Illumina HiSeq4000 (human) or NovaSeq6000 (murine) using paired-end as sequencing mode.

**Single-cell RNA sequencing data analyses**. Single-cell expression data were processed using the Cell Ranger Single Cell Software Suite (v2.1.1) and StarSolo to perform quality control, sample de-multiplexing, barcode processing, and single-cell 3′ gene counting[26]. Sequencing reads were aligned to the human reference genome GRCh38 using the Cell Ranger suite with default parameters. Dimensional reduction analysis was performed in Seurat (v3) R (v3.6)[27]. Monocle (v2.6.0) was applied for differential expression analysis and the generation of cell trajectories[28]. The gene-cell-barcode matrix of the samples was log-transformed and filtered based on the number of genes detected per cell (any cell with less than 200 genes per cell or greater than 10% mitochondrial content was filtered out). Regression in gene expression was performed based on the number of UMI. PCA was run on the normalized gene-barcode matrix. Barnes-hut approximation to t-SNE was then performed on principal components to visualize cells in a two-dimensional space[29]. This graph-based clustering method relies on a clustering algorithm based on shared nearest neighbor (SNN) modularity optimization. Differential transcriptional profiles by cluster were generated in Seurat with associated GO terms derived from the functional annotation tool DAVID Bioinformatics Resources 6.7 (NIAID/NIH, https://david.ncifcrf.gov/summary.jsp) and Metascape 3.5. Cell annotation was then performed by assessing the relative expression of standard endothelial and hematopoietic markers. Cell fate trajectory analysis (Monocle) was utilized on an

EMCN enriched cluster of cells to identify unique attributes between heart failure and healthy patients' cells.

**HUVEC cell culture**. The cell culture assays were performed with human umbilical vein ECs (HUVECs). HUVECs were purchased from Lonza. Cells were cultured in EC basal medium (EBM; CC-3121; Lonza), supplemented with EGM-SingleQuots (CC-4133; Lonza), and 10% fetal bovine serum (10270-106; Gibco). ECs were kept at 37 °C and 5% CO₂. For cytokine assays, cells were stimulated with IL-1β (100 ng/mL; 201-LB; R&D Systems) 24 h after seeding for 1 h.

**Liver EC isolation**. To analyze the expression levels of human MYC in the $Myc^{EC-OE}$ mice, ECs from the liver were isolated as in Glaser et al.[30]. After perfusion with GBSS (G9779; Sigma), the liver was dissociated with collagenase type-II (17101-015; Gibco; 600 U/mice). The dissociated tissue was transferred to C-tubes and mechanically shred (808426759; gentleMACS™ Dissociator; Miltenyi). The cells were filtered with a 200 µm pore strainer (43-50200-03; Pluri select) and separated using Nycodenz solution (35%; 1002424; AXIS-SHIELD). The ECs were isolated using LSEC-Beads (130092007; Miltenyi) for 30 min at 4 °C. LS-column (130-042-401; Miltenyi) was used for magnetic separation. The EC-positive fraction was dissolved in QIAzol and RNA isolated.

**Cell RNA isolation and RT-qPCR**. To analyze the expression levels of MYC in stimulated HUVECs or in isolated murine liver ECs, cells were lysed with Qiazol Lysis Reagent (79306; Qiagen). The RNA was isolated using the miRNeasy-kit (217004; Qiagen) with additional DNase-I digestion (79254; Qiagen) according to the manufacture's protocol. The concentration of the isolated RNA was measured with the NanoDrop2000 spectrophotometer (Thermofisher Scientific). In total, 1000 ng of RNA were reverse-transcribed using the MMLV Transcriptase (N8080018; Life technologies) and random hexamer primer (N808-0261; Life technologies). The reverse-transcribed RNA was then used for quantitative real-time PCR analysis on a ViiA7-Real-time qPCR System (Life Technologies), using Fast SYBR Green Mastermix (#4385612 Life Technologies). Primer sequences can be found on Supplementary Table 2. mRNA levels of human MYC in murine liver ECs (forward: CGGATTCTCTGCTCTCCTCG, reverse: TCATCTTCTTGTTCCT CCTCAGA) and human MYC in HUVECs (forward: AGCTGCTTAGACGCTG GATTTT, reverse: TCGAGGTCATAGTTCCTGTTGG) were normalized to murine Rplp0 (forward: TTTGACAACGGCAGCATTTA, reverse: CCGATCTGC AGACACACACT) or human RPLP0 (forward: GGCGACCTGGAAGTCCAACT, reverse: CCATCAGCACCACAGCCTTC), using the $2^{-\Delta Ct}$ method.

**MYC transcription factor assay**. To analyze the activation of MYC after IL-1β (100 ng/mL; 201-LB; R&D Systems) treatment, Myc Transcription Factor Assay Kit (ab207200; abcam) was used, according to the manufacture's protocol. The nuclear extracts were isolated using the nuclear extraction kit (ab113474; abcam), according to the manufacturer's protocol for adherent cells with the detachment step using Trypsin-EDTA (15400054; Thermo Fisher Scientific) for 3 min at 37 °C. The protein concentration was measured using the Bradford method (798 µl Bradford (500-0006; BioRad), 200 µl water, and 2 µl of the protein) at a wavelength of 595 nm (SmartSpecPlus; BioRad). For the transcription factor assay, 5 µg of the isolated nuclear extracts were used and the absorbance was measured on a spectrophotometer (Synergy HT; BioTek) at OD 450 nm.

**Caspase I activation**. Pyroptosis in HUVECs was measured using FAM Caspase-1 Kit based on FLICA technology to detect caspase-1 activation (Bio-Rad Laboratories GmbH; ICT9146). FAM–VAD–FMK was dissolved in dimethyl sulfoxide (DMSO; Sigma) following the manufacturer's instructions. HUVEC cells were seeded in 6 well cell culture plates (657160; Greiner Bio-One) and stimulated after 24 h with Nigericin (10 µM; N7143; Sigma Aldrich) or IL-1β (100 ng/mL; 201-LB; R&D Systems) for 3 h at

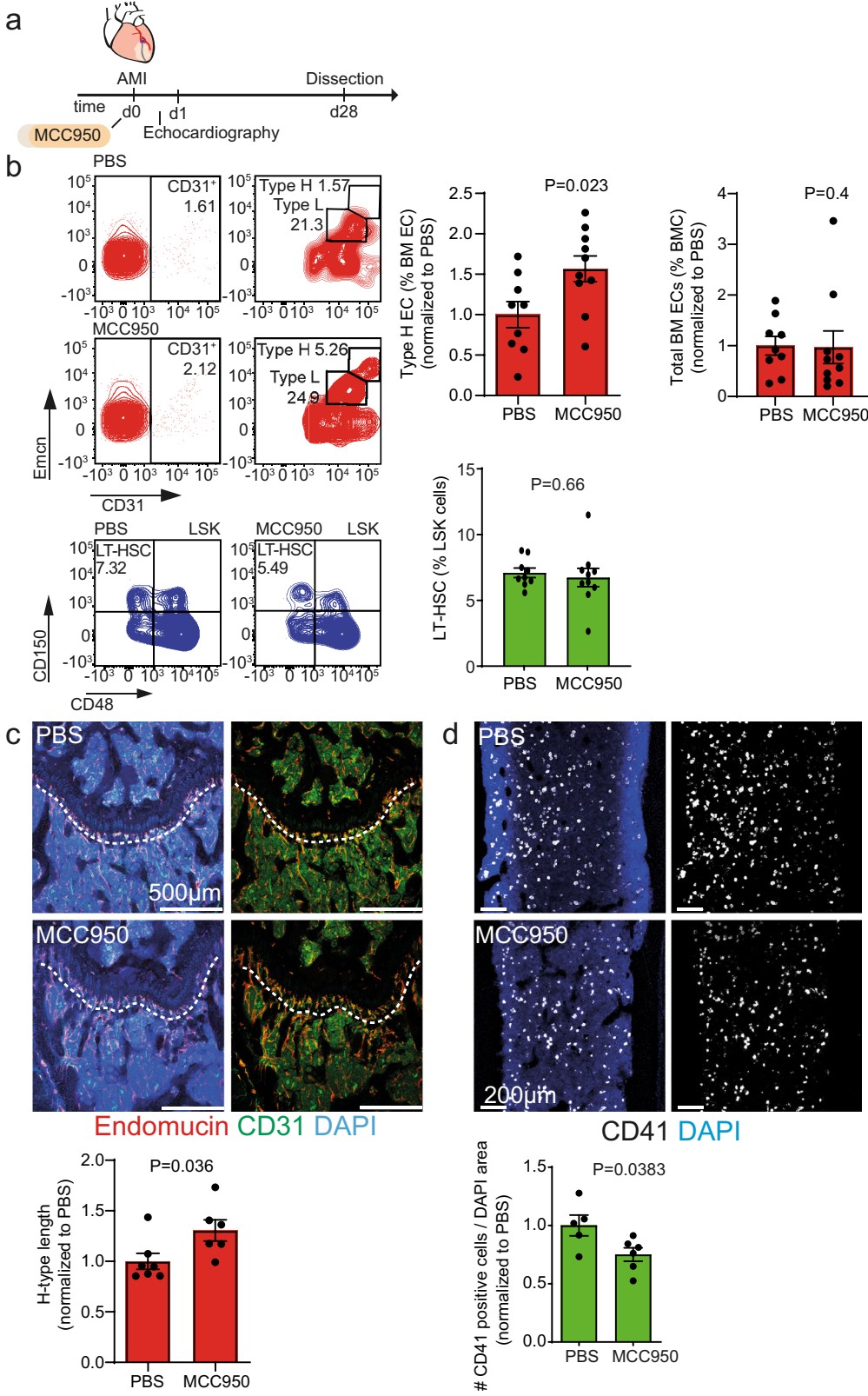

37 °C. HUVEC were detached with Accutase® Cell Detachment Solution (A6964-100ML; Sigma Aldrich), centrifuged (500×g, 5 min), and resuspended in HUVEC medium. FLICA staining was then prepared by adding 5 µl of FLICA working solution into 295 µl of culture medium. Cells were incubated for 30 min at 37 °C. Following two washing steps, HUVECs were resuspended in 500 µl 1× washing buffer and stained with 5 µL 7AAD (420404; Biolegend), prior to acquisition on BD FACS Canto II flow cytometer. The data were analyzed using FlowJo (Version 10; FlowJo LLC).

**Statistics**. GraphPad's 9 (Prism) software was used for statistical analysis of all experiments but scRNA-seq. Shapiro–Wilk normality test was used to test data before comparison. Unpaired, two-tailed Student's *t*-test and Mann–Whitney test were used for comparison between two groups. For multiple comparisons, one-way ANOVA with Dunnet's multiple comparisons test was used. Data is presented in scatter plots with mean ± standard error of the mean (sem). Differences were considered statistically significant at $p < 0.05$.

**Fig. 4 Anti-IL-1β treatment ameliorates the loss of type H vasculature in the bone after MI. a** Schematic of the experimental design. **b** Flow cytometry analysis of femur bone marrow. Left, gating strategy; right, quantification. The numbers shown within the gates represent the percentages of events (cells) within that gates relative to the corresponding parent gate. The numbers shown within the gates represent the percentages of events (cells) within that gates relative to the upstream parent gates (CD45$^{neg}$Ter119$^{neg}$ for CD31$^{pos}$ cells and CD31$^{pos}$ cells for Type H and Type L EC subsets, accordingly) (red). Type H endothelial cell number is increased upon anti-IL-1β treatment (MCC950), while the total number of endothelial cells is not affected. $N = 9$ for PBS and $N = 10$ for MCC950. Data are shown as mean ± SEM, normalized to PBS. $P$-value was calculated by unpaired, two-tailed Student's $t$-test (type H endothelial cell number) and two-tailed Mann–Whitney test (total number of endothelial cells). EC,# endothelial cells, BM EC, bone marrow endothelial cells (green). The total number of long-term hematopoietic stem cells (LT-HSC) remains unchanged after IL-1β treatment. $N = 9$ PBS and $N = 10$ MCC950. Data are shown as mean ± SEM. $P$-value was calculated by unpaired, two-tailed Student's $t$-test. LSK, Lin$^-$Sca-1$^+$c-kit$^+$ cells (**c**) and (**d**) Immunostaining of longitudinal femur sections. **c** The length of type H vessels (dashed line) is longer upon MCC950 treatment 28 days after MI. $N = 7$ for PBS and $N = 6$ for MCC950. Data are shown as mean ± SEM. $P$-value was calculated by unpaired, two-tailed Student's $t$-test. **d** CD41$^+$ myeloid progenitor cell number is decreased in the bone marrow of MCC950-treated animals after myocardial infarction. $N = 5$ PBS and $N = 6$ MCC950. Data are shown as mean ± SEM. $P$-value was calculated by unpaired, two-tailed Student's $t$-test.

**Reporting summary**. Further information on research design is available in the Nature Research Reporting Summary linked to this article.

## Data availability

The single-cell RNA-seq data sets generated in this study are available at Array Express (https://www.ebi.ac.uk/arrayexpress) with the following accession numbers: E-MTAB-10432 and E-MTAB-10448. All other data are included within the article, source data, and supplementary data can be made available upon request. Source data are provided with this paper.

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

## Acknowledgements

We thank Eva-Maria Rogg, Bianca Schuhmacher, and Marga Müller-Ardogan for excellent technical support and patient care and Halvard Bönig for support in establishing and validating human flow cytometry panels. The study was supported by the German Research Foundation (SFB 834; project B6 to B.A., J.H., and A.M.Z.; project B1 to S.D.), the German Center for Cardiovascular Research, Berlin, Germany, to S.D. and A.M.Z. and the Dr. Rolf M. Schwiete Stiftung to S.D. The work by M.P. was supported by the European Research Council Consolidator Grant EMERGE (773047).

## Author contributions

J.H., Conceptualization, Data curation, Formal analysis, Investigation, Funding acquisition, Writing—original draft. G.L., Conceptualization, Data curation, Formal analysis, Investigation, Writing—original draft. W.T.A., Conceptualization, Data curation, Formal analysis, Investigation, Writing. S.F.G. Investigation, Methodology. T.R., Investigation, Methodology. A.F., Investigation, Methodology. M.M.R., Investigation, Methodology. D.J., Investigation, Methodology. M.P., Methodology, Provided transgenic mouse lines. B.A., Investigation, Methodology, Patient recruitment, Funding acquisition. A.M.Z., Conceptualization, Supervision, Funding acquisition, Project administration. S.D., Conceptualization, Supervision, Funding acquisition, Writing—original draft, Project administration

## Funding

## Competing interests

The authors declare no competing interests.
