## [Peer Review File · Nature Communications]

Reviewers' comments:

Reviewer #1 (Remarks to the Author):

In this manuscript, Dr. Hoffman and colleagues examined how MI affects the bone vascular niche in mice and humans. They found a loss of type H bone endothelium after MI. I have a few comments aimed at improving this submission.

1. For the reduction in type H cells, the results text discusses MI d7 while the representative in figure 1a shows d3. Why are different times discussed and shown?
2. The change in type H cells and LT-HSCs and the development of HF could be coincidental. It is not clear if the authors are claiming the loss in type H cells with MI is a cause of later HF.
3. How do you define post-infarction HF in mice? The classical definition of EF <45% would apply to mice after 24 hours of MI.
4. For the human scRNA experiment, the 2 groups have n=1 each. How was statistics performed such that so many conclusions can be drawn as being significant? This data appear to be way over analyzed.
5. The term strikingly to describe an increase in IL-1b in type H cells after MI is strange- IL-1b is a predominant cytokine at MI d1.
6. Direct cause and effect linkages have not been made.
7. Figure 2 legend- what does mainly populated mean?
8. Fig 3a- it is interesting that IL-1b mRNA is not increased until day 28 after MI, but the protein is increased at day 1. Are there PMNs present to account for the mismatch between mRNA and protein?
9. Supp Fig 1 is not needed- would be better to have a supplemental table of echo measurements, along with infarct size.

Reviewer #2 (Remarks to the Author):

In the following work Hoffman et al. examine effect of myocardial infarction (MI) on bone marrow (BM) vasculature and hematopoiesis. Heart failure resulted in a loss of H-vessels in the BM and skewed hematopoiesis. scRNA-seq analysis identified an upregulation of inflammatory genes in the bone marrow H vessels of a patient with post MI-heart failure. Interestingly, blocking inflammation partially protected BM H-vessels from the effect of myocardial infarction.

Overall, this is a very interesting work, examining relationship between cardiac function, bone marrow niche and hematopoiesis. The findings have a significant clinical relevance and is of interest to a broad scientific audience. The authors demonstrate increased frequencies of LT-HSCs and CD41 staining of BM sections, but that alone is not sufficient to understand the impact of hematopoiesis. It is critical that the authors perform a thorough hematopoietic analysis follow MI inductions +/- MCC950 treatment (see specific comments). My second concern is scRNA-seq analysis. It is a very granular analysis, yet it is not clear to me which pathways are enriched in H-vessel of HF patient. Given IL1b upregulation, can the authors detect an enrichment in inflammatory pathways? Are there cell cycle differences between the patients? What are gene signatures of H-vessels? What are differentially expressed genes between the patients? The authors should make the single-cell data available to referees during the revision.

Specific comments:

- Hematopoietic analysis: Cell frequencies should be indicated for all flow cytometry gates.

Frequencies and absolute numbers should be provided for all populations analyzed. Please refer to Herault et al 2017 (Fig .1c and SupFig.3b). Analysis of LSKs, LT-HSC, MPP3/4, GMP, and Gr should be performed to properly assess myeloid skewing.

- Fig. 1c, 3d: High resolution images should be used. It is very hard to see. Also, the authors should provide representative images of both metaphysis and diaphysis.

- Fig. 3: IL1b concentration in BM serum should be measure by ELISA.

- Fig.3: Please provide schematic of experiment, indicating when MCC950 was administered.

Reviewer #3 (Remarks to the Author):

The authors present a well-conceived and stimulating investigation of the interaction between the diseased heart and bone marrow. Their hypothesis that myocardial infarction itself leads to decreased Endomucin+CD31+ type H endothelium in the bone marrow and myeloid bias through IL1-B and the NRLP3 inflammasome is by and large well-supported in the mouse data at 3 days and especially at 28 days after LAD ligation. Similar correlations were demonstrated in patients with heart failure.

Comments:

Murine Studies:

1) Most of the data presented in figure 1 report changes in the Endomucin+ Type H endothelial cells in the bone marrow in terms percent of the cells but not the total number of the cells present in the bone marrow. While the authors present data of the total number of endothelial cells, it is informative that such data be presented for the challenged mouse bone marrow as well. Could it be that rather than the total number of Type H endothelial cell being depleted, there is a downregulation of the expression of Endomucin or CD31 markers?

2) The increase in the total number of LT-HSCs after induction of infarction is intriguing. Since the majority of the data in the quantification of LT-HSCs is based on phenotypic quantification, it is imperative that the authors provide a more broad information on the profile of the other hematopoietic lineages, as well. What causes an increase in LT-HSCs? What happens to the lineage differentiated cells? Could this represent a manifestation of "Clonal Hematopoiesis"?

3) The mechanism by which Type H endothelial cells undergo pyroptosis is unclear. The stainings demonstrating the loss of Type H endothelial cells is too preliminary to draw those conclusions. The data generated from the inhibitor of inflammasome, MCC950 is mostly correlative.

4) The authors acknowledge that signals from the injured myocardium is relayed to the bone marrow causing Type H endothelial cell loss. At minimum the authors might consider demonstrating whether this is mediated via humoral factors or through cellular activation of circulating cells. What is the biological significance of pyroptosis of Type H endothelial cells as compared to Type L endothelial cells.

5) It is unclear whether the control mice received sham thoracotomies without LAD ligation. If not, then it would be difficult to ascribe the post-ligation day 3 alterations in the bone marrow to the MI alone, since the post-surgical inflammatory state would also contribute. However, the gradually increased effect on the bone marrow by 28-day data argues in favor of the authors' conclusions and mitigates this concern.

Human studies:

1) The authors contend that the scRNA analysis of 1 healthy and 1 ischemic heart failure patient support the relevance of the murine findings to human. Strictly speaking, this is true, but it does

not provide robust support, since two adult humans of similar and presumably advanced age may have vastly different clinical backgrounds that give unknown confounders to findings in the blood and bone marrow (i.e. diabetes, smoking, RA, and any inflammatory condition). This could be added to supplementary 1 along side the HF patient with scRNA data.

2) Likewise, no clinical data is given for the 8 "healthy" in the FACS analysis other than their median age and that they had no evidence of CAD. This data could also be added to supplementary table 1. The fact that their median age being about 1/2 that of the heart failure patients is also concerning and might not be statistically sufficient to establish that the type H depletion is age-independent in humans as clearly shown in mice. Given the difficulty of obtaining ideal controls through this challenging procedure, the reviewer feels this is an acceptable limitation in the study, but should be acknowledged in the text or tables.

Minor points:

- The final paragraph regarding the link between heart failure and fractures is supported only by retrospective data in *Circulation*, which could be confounded by low-activity in HF individuals/ low sunlight. Since the paper does not directly address bone strength in human or mouse data, it might be stronger without this as the final point in the conclusion.
- 17 out of 18 humans were male. All mice were male. The male/female split of the healthy controls is not given. While CAD/ischemic HF does show is predominantly found in males, this seems appropriate, but please add the gender of the controls as well since estrogen is known to have its own protective effects on endothelium.

Reviewer #1

In this manuscript, Dr. Hoffmann and colleagues examined how MI affects the bone vascular niche in mice and humans. They found a loss of type H bone endothelium after MI. I have a few comments aimed at improving this submission.

We thank the reviewer for the following comments, which we all addressed in the revised manuscript.

1. For the reduction in type H cells, the results text discusses MI d7 while the representative in figure 1a shows d3. Why are different times discussed and shown?

We have replaced the day-3 plot with the control plot so that the figure matches the text. The text now reads as follows:

Type H cells were significantly decreased by day 28 as compared to control mice (Fig. 1b).

2. The change in type H cells and LT-HSCs and the development of HF could be coincidental. It is not clear if the authors are claiming the loss in type H cells with MI is a cause of later HF.

We have addressed this concern by the reviewer by analyzing the mechanistic relationship between IL-1b, cMYC and the loss of the H type vasculature. We have observed that IL-1b is sufficient to induce the expression and activation of cMYC in endothelial cells, but more interestingly that the EC-specific overexpression of cMYC, in an infarct free environment is sufficient to cause a reduction of the H type vasculature. This is now represented by new Figure 3 and it appears in text as:

we tested whether overexpression of MYC is sufficient to induce a loss of type H cell in vivo. To do so, we bred the R26StopFLMYC mouse line that bears a human MYC cDNA preceded by a loxP-flanked “stop” cassette in the Rosa26 locus with Cdh5CreERT2 transgenic animals expressing tamoxifen-inducible Cre recombinase specially in ECs⁹ (Fig. 3h). Endothelial-specific overexpression of human MYC in MycEC-OE mutant mice, indeed induced a significant reduction of H type endothelium (Fig. 3i).

3. How do you define post-infarction HF in mice? The classical definition of EF < 45 % would apply to mice after 24 hours of MI.

We have update Supplementary figure 1 to show not only the EF measurements in the first 24h but also a representative image of the infarcts as well as a quantification of the infarct sizes at d28.

4. For the human scRNA experiment, the 2 groups have n=1 each. How was statistics performed such that so many conclusions can be drawn as being significant? This data appears to be way over analyzed.

The statistical test is performed on measured events (cells) for the relative mRNA expression levels, which is customary in such experiments. Given the high cost of single cell RNA sequencing experiments and rarity of the samples for human bone marrow from aged matched control and post-infarctions heart failure, these data should be considered as a screening and hypothesis-generating approach.

To substantiate our findings, we have performed single-cell RNA sequencing in post-MI mice to confirm the human study. This experiment was performed in $n=9$ mice: 3 pooled control mice, 3 pooled d7 mice after MI and 3 pooled d28 mice after MI. This experiment confirmed the reduction of the Endomucin rich endothelial population accompanied by an increase of inflammation signals. The results of this analysis appear on new Figure 3 and new Supplementary figure 7 and read in the main text as follows:

Next, we validated the human data by assessing the impact of MI on the transcriptome of the murine bone marrow vascular niche at the single cell level (Fig. 3a and Supplementary Fig. 7a, b). Endomucin expression was significantly reduced at d28 (Fig. 3b) while the expression of *Il1b* and *Myc* are significantly increased post-MI (Fig. 3c). The induction of *IL-1 β* protein in type H cells post-MI was confirmed by immunostainings of bone sections (Fig. 3e and Supplementary Fig. 7c). Gene ontology (GO) analysis of the upregulated genes in the murine endothelial cells at d7 post-MI relative to d0 confirmed the inflammatory response (Fig. 3d). Further analysis at d28 relative to d0 revealed suggested increased cell stress and cell death, indicated by enriched GO terms like “Cellular response to stress”, “Autophagy” or “Positive regulation of cell death” (Supplementary Fig. 7b). Furthermore, inflammation-related transcripts had a prevailing impact on gene expression signatures (e.g. “Adaptive immune system” or “Leukocyte migration”) at d28 (Supplementary Fig. 7b).

5. The term strikingly to describe an increase in IL-1b in type H cells after MI is strange IL-1b is a predominant cytokine at MI d1.

We have tone down this claim as suggested by the reviewer, the new sentence reads as follows:

The induction of *IL-1 β* protein in type H cells post-MI was confirmed by immunostainings of bone sections (Fig. 3e and Supplementary Fig. 7c).

6. Direct cause and effect linkages have not been made.

To study the direct cause and effect relationship, we analyzed the involvement of endothelial *cMyc* first. As already described in the response to point number 2, *IL-1 β* induces the expression and the activation of *cMYC* in endothelial cells. Furthermore, the overexpression of *cMYC* in endothelial cells *in vivo* is sufficient, in an infarct free environment, to induce the loss of H type vasculature. This is shown in new figure 3.

In addition we tested the role of adrenergic signaling, which is known to be activated upon MI and was shown to affect the bone marrow stem cell niche. However, the β 2AR antagonist ICI-118,551 did not prevent infarction-induced loss of H type vasculature or expansion of the myeloid population. The results of this experiment are now present in new Supplementary figure 8 and 9 and read in the text as follows:

Previous studies showed that MI-associated sympathetic activity and stimulation of the bone niche drives an inflammation-mediated deterioration of the niche^{10,11}. Moreover, increased β 2-adrenergic activity has been shown to substantially influence niche microenvironment by promoting IL-6-dependent CD41+ myeloid progenitor expansion¹⁰. We therefore speculated that blocking beta-2 adrenergic receptor (β 2AR) signalling protects the vascular niche in the bone after MI. We performed MI in the presence of ICI-118,551,12,13, a well-known β 2AR antagonist (Supplementary Fig. 8a and Supplementary Fig. 9). However, we could not detect effects on MI-induced reduction of H type endothelium demonstrating that activation of β 2AR signaling does not mediate deterioration of the bone vascular niche (Supplementary Fig. 8b, d) or expansion of the myeloid progenitors (Supplementary Fig. 8c, e). This excludes the contribution of sympathetic humoral signalling to the reduction of H type vasculature after MI.

Together our data demonstrate that inflammatory signals involving the inflammasome activation but not neurohumoral activation mediate the effect of MI on the BM vascular niche. Moreover, the causal involvement of MYC was demonstrated in two ways. First we have shown that IL-1 β induces not only the expression of MYC but also the activation of the protein and second, we have demonstrated that the overexpression of MYC specifically in endothelial cells is sufficient to cause H type vasculature loss in the bone.

7. Figure 2 legend- what does mainly populated mean?

We have rephrased the figure legend to make it more clear. Now it reads as follows:

“(h) Distribution of cells among pseudotime states and relative IL1B expression. Distribution analysis revealed that states 10, 11, 12, and 13 consist mainly of HF patient cells. IL1B expression is higher in states 12 and 13. Dashed line indicates normalized Unique Molecular Identifier (nUMI) counts of 2.5.”

8. Fig 3a- it is interesting that IL-1b mRNA is not increased until day 28 after MI, but the protein is increased at day 1. Are there PMNs present to account for the mismatch between mRNA and protein?

We agree that the discrepancy might relate to the enrichment of non-endothelial cells during the time course after MI. We therefore, instead of whole bone RT-qPCR, performed single cell RNA sequencing studies to determine the expression of IL-1 β specifically in bone marrow endothelial cells. This analysis shows a significant induction of IL-1 β at day 7 to day 28 (new Figure 3 and Supplementary Fig. 7)

9. Supp Fig 1 is not needed- would be better to have a supplemental table of echo measurements, along with infarct size.

We have changed Supplementary figure 1 by a table that shows EF measurements in the 24h and infarct size at d28 along with representative figures of the infarct size after 28 days.

Response to Reviewer #2

In the following work Hoffmann et al. examine effect of myocardial infarction (MI) on bone marrow (BM) vasculature and hematopoiesis. Heart failure resulted in a loss of H-vessels in the BM and skewed hematopoiesis. scRNA-seq analysis identified an upregulation of inflammatory genes in the bone marrow H vessels of a patient with post MI-heart failure. Interestingly, blocking inflammation partially protected BM H-vessels from the effect of myocardial infarction.

Overall, this is a very interesting work, examining relationship between cardiac function, bone marrow niche and hematopoiesis. The findings have a significant clinical relevance and is of interest to a broad scientific audience. The authors demonstrate increased frequencies of LT-HSCs and CD41 staining of BM sections, but that alone is not sufficient to understand the impact of hematopoiesis. It is critical that the authors perform a thorough hematopoietic analysis follow MI inductions +/- MCC950 treatment (see specific comments). My second concern is scRNA-seq analysis. It is a very granular analysis, yet it is not clear to me which pathways are enriched in H-vessel of HF patient. Given IL1b upregulation, can the authors detect an enrichment in inflammatory pathways? Are there cell cycle differences between the patients? What are gene signatures of H-vessels? What are differentially expressed genes between the patients? The authors should make the single-cell data available to referees during the revision.

We thank the reviewer for his positive comments acknowledging the clinical relevance of this study. By addressing the specific comments, we believe that we have improved the manuscript. The single cell RNA sequencing is now available for the reviewers during the revision. Please find the original Seurat file under the following link:

<https://figshare.com/s/ad850df5d7643186c965>.

In addition, we have provided GO terms indicating enrichment of inflammatory pathways in the scRNA-seq in both human and mouse models in the new manuscript (Fig. 3 and Supplementary Fig 7).

Specific comments:

1. Hematopoietic analysis: Cell frequencies should be indicated for all flow cytometry gates. Frequencies and absolute numbers should be provided for all populations analyzed. Please refer to Herault et al. 2017 (Fig .1c and SupFig. 3b).

We have performed the requested analysis and included cell frequencies gates in our FACS gates and absolute cell numbers in the extended hematopoietic flow cytometry panels present in supplementary figure 4. The method section is up-dated and cites the publication referred to by the reviewer.

2. Analysis of LSKs, LT-HSC, MPP3/4, GMP, and Gr should be performed to properly assess myeloid skewing.

We have extended our hematopoietic flow cytometry panels to more thoroughly assess myeloid skewing. The results are present in the Figure for the reviewer (see below) and in Supplementary Figure 4. The main text now reads as follows:

To understand the additional impact on hematopoietic cells in the post-MI bones, we performed a more detailed flow cytometric analysis of the bone haematopoietic stem and progenitor cell compartments (Supplementary Fig. 4a). This analysis revealed a significant expansion of myeloid-biased progenitors, as indicated by the increased CD41 expression at day 28 post-MI (Supplementary Fig. 4b, c).

3. Fig. 1c, 3d: High resolution images should be used. It is very hard to see. Also, the authors should provide representative images of both metaphysis and diaphysis.

We have corrected the figures by adding high resolution panels. Also we have added representative images of the diaphysis corresponding to Fig. 1c and 3d. These images appear in Supplementary figures 2, 8 and 9. In case of Fig1c, these images are referred to in the text as follows:

whereas Type L cells in the diaphysis of the bone were not changed post MI (Supplementary Fig. 2).

4. Fig. 3: IL1b concentration in BM serum should be measure by ELISA.

Since reviewer #1 was concerned that non-endothelial cells may confound the results achieved when using total bone marrow, we have endeavored to go one step further and assessed the differences at single cell level. We have seen enhanced response to cytokine signaling in post-infarct murine bone marrow cells at single cell resolution. These results are present in Fig. 3c,d and Supplementary Fig. 7b. These observations support our hypothesis that the phenotype observed in the bone vascular niche is influenced by circulating inflammatory signals.

Next, we validated the human data by assessing the impact of MI on the transcriptome of the murine bone marrow vascular niche at the single cell level (Fig. 3a and Supplementary Fig. 7a, b). Endomucin expression was significantly reduced at d28 (Fig. 3b) while the expression of Il1b and Myc are significantly increased after d7 post-MI (Fig. 3c). The induction of IL-1 β protein in type H cells post-MI was confirmed by immunostainings of bone sections (Fig. 3e and Supplementary Fig. 7c). Gene ontology (GO) analysis of the upregulated genes in the murine endothelial cells at d7 post-MI relative to d0 confirmed the inflammatory response (Fig. 3d). Further analysis at d28 relative to d0 revealed suggested increased cell stress and cell death, indicated by enriched GO terms like “Cellular response to stress”, “Autophagy” or “Positive regulation of cell death” (Supplementary Fig. 7b). Furthermore, inflammation-related transcripts had a prevailing impact on gene expression signatures (e.g. “Adaptive immune system” or “Leukocyte migration”) at d28 (Supplementary Fig. 7b).

5. Fig.3: Please provide schematic of experiment, indicating when MCC950 was administered.

MCC950 and ICI-118,551 pumps were implanted at d0 taking advantage that the animals were under anesthesia. We have added schematic of all the experiments to the corresponding figures. Furthermore, we have updated the methods part to make this clearer. Now the methods read as follows:

“Mini-osmotic pumps were implanted in the same intervention where MI was induced”

Reviewer #3

The authors present a well-conceived and stimulating investigation of the interaction between the diseased heart and bone marrow. Their hypothesis that myocardial infarction itself leads to decreased Endomucin+CD31+ type H endothelium in the bone marrow and myeloid bias through IL1-B and the NRLP3 inflammasome is by and large well-supported in the mouse data at 3 days and especially at 28 days after LAD ligation. Similar correlations were demonstrated in patients with heart failure.

We thank the reviewer for his positive comments highlighting how stimulating this study is.

Comments:

Murine Studies:

1. Most of the data presented in figure 1 report changes in the Endomucin+ Type H endothelial cells in the bone marrow in terms percent of the cells but not the total number of the cells present in the bone marrow. While the authors present data of the total number of endothelial cells, it is informative that such data be presented for the challenged mouse bone marrow as well. Could it be that rather than the total number of Type H endothelial cell being depleted, there is a downregulation of the expression of Endomucin or CD31 markers?

The immunostainings (Figure 1c) of the bone metaphysis show that the type H endothelial cells gradually disappear after MI indicated by measurement of the length of the vessels expressing high levels of Endomucin and CD31. Despite being shorter, these vessels still show a high expression of both markers. Leftover H type endothelial cells still express modestly high levels of Endomucin. This can be observed in the post-MI mouse single-cell-RNA-sequencing in Figure 3a and 3b. The reduction in the average gene expression detected by scRNAseq is significantly influenced by the decline in the numbers of Emcn expressing cells.

2. The increase in the total number of LT-HSCs after induction of infarction is intriguing. Since the majority of the data in the quantification of LT-HSCs is based on phenotypic quantification, it is imperative that the authors provide a broader information on the profile of the other hematopoietic lineages, as well. What causes an increase in LT-HSCs? What happens to the lineage differentiated cells? Could this represent a manifestation of "Clonal Hematopoiesis"?

To address the questions of the reviewer, we performed the following experiments: we assessed other hematopoietic lineages by applying extended flow cytometry panels. This analysis revealed the significant expansion of myeloid-biased progenitors, as indicated by the increased CD41 expression and cell numbers at day 28 post MI (Supplementary Fig. 4b, c). In addition, we could show a significant expansion of granulocyte-monocyte myeloid progenitor (GM-P) and megakaryocyte progenitor (MkP) populations over time. These results are represented in the new Supplementary Fig. 4 and in the figure to the reviewers present above. The text reads as follows:

To understand the additional impact on hematopoietic cells in the post-MI bones, we performed a more detailed flow cytometric analysis of the bone haematopoietic stem and progenitor cell compartments (Supplementary Fig. 4a). This analysis revealed a significant expansion of myeloid-biased progenitors, as indicated by the increased CD41 expression at day 28 post-MI (Supplementary Fig. 4b, c).

In addition, we discussed the potential relation of these findings to clonal hematopoiesis in the text of the revised manuscript as follows:

Since MI can drive myeloid skewing and LT-HSC numbers, it is interesting to speculate whether MI may expedite expansion of HSCs harbouring CHIP-driver mutations. Indeed, in cohorts of patients with ischemic heart failure, it was found that these cohorts have a higher incidence of patients harbouring CHIP-driver mutations relative to similar age matched cohorts published elsewhere^{20,21}. Interestingly, an exploratory analysis of patients harbouring CHIP mutations (TET2) from the CANTOS trial demonstrated protection from death and hospitalization greater than the non-CHIP population²².

3. The mechanism by which Type H endothelial cells undergo pyroptosis is unclear. The stainings demonstrating the loss of Type H endothelial cells is too preliminary to draw those conclusions. The data generated from the inhibitor of inflammasome, MCC950 is mostly correlative.

To gain further insights into the mechanism, we first used in vitro studies confirming that IL-1 β induces MYC and activates pyroptosis. These data can be seen in figures 3 and supplementary figure 8 and reads in the text as follows:

we used cultured endothelial cells to determine whether IL-1 β regulates MYC. Indeed, IL-1 β induced an upregulation of the MYC transcript and an increase in the activation of the protein as measured by ELISA in cultured endothelial cells (Fig. 3f). Furthermore, IL-1 β treatment induced pyroptosis in endothelial cells indicated by caspase-1 activation by flow cytometry (Fig. 3g)

In addition, we demonstrate that overexpression of MYC, specifically in endothelial cells, induces H type vasculature loss in vivo. This data is represented in the new figure 3 and read in the text as follows:

we tested whether overexpression of MYC is sufficient to induce a loss of type H cell in vivo. To do so, we bred the R26StopFLMYC mouse line that bears a human MYC cDNA preceded by a loxP-flanked “stop” cassette in the Rosa26 locus with Cdh5CreERT2 transgenic animals expressing tamoxifen-inducible Cre recombinase specially in ECs⁹ (Fig. 3h). Endothelial-specific overexpression of human MYC in MycEC-OE mutant mice, indeed induced a significant reduction of H type endothelium (Fig. 3i).

We have further shown that that inhibition of inflammasome prevents MI-induced loss of Type H cells and induction of myeloid cell. We exclude a role of adrenergic stimulation. To do so, we tested the role of adrenergic signaling, which is known to be activated upon MI and was shown to affect the bone marrow stem cell niche. However, the β 2AR antagonist ICI 118,551 did not prevent infarction induces loss of H type vasculature reduction or of expansion of the myeloid population. The results of this experiment are now present on new Supplementary figure 8 and read in the text as follows:

Previous studies showed that MI-associated sympathetic activity and stimulation of the bone niche drives an inflammation-mediated deterioration of the niche^{10,11}. Moreover, increased β 2-adrenergic activity has been shown to substantially influence niche microenvironment by promoting IL-6-dependent CD41+ myeloid progenitor expansion¹⁰. We therefore speculated that blocking beta-2 adrenergic receptor (β 2AR) signalling protects the vascular niche in the bone after MI. We performed MI in the presence of ICI-118,551^{12,13}, a well-known β 2AR antagonist

(Supplementary Fig. 8a and Supplementary Fig. 9). However, we could not detect effects on MI-induced reduction of H type endothelium demonstrating that activation of β 2AR signaling does not mediate deterioration of the bone vascular niche (Supplementary Fig. 8b, d) or expansion of the myeloid progenitors (Supplementary Fig. 8c,e). This excludes the contribution of sympathetic humoral signalling to the reduction of H type vasculature after MI.

Together our observations demonstrate that inflammatory signals involving the inflammasome activation but not neurohumoral activation mediate the effect of MI in the bone vascular niche. We have demonstrated that MI induces the loss of H type vasculature in a cMYC dependent manner in two ways. First we have shown that IL-1 β not only induces the expression of cMYC but also promotes the activation of the protein and induces pyroptosis in endothelial cells and second, we have demonstrated that the overexpression of cMYC specifically in endothelial cells is sufficient to cause H type vasculature loss in the bone. The text discussing this mechanism reads as follows:

Our studies further suggest a crucial role of IL1 β -mediated activation of MYC in the H type vasculature. IL-1 β induced MYC expression and is sufficient to cause pyroptosis in endothelial cells in vitro. More interestingly, forced expression of MYC can induce a loss of H type vessels in the absence of inflammatory signals. Since MYC has been linked to cellular death^{7,8}, we speculate that inflammatory activation of CD31^{hi}EMCN^{hi} endothelial cells induce pyroptotic cell death leading to the accelerated depletion of of type H endothelium. This might trigger an age-independent disruption of endothelium instructive function in the bone, leading to dysregulation of haematopoiesis and HSC activity, marked by expansion and myeloid lineage skewing of HSCs^{15,16}.

4. The authors acknowledge that signals from the injured myocardium is relayed to the bone marrow causing Type H endothelial cell loss. At minimum the authors might consider demonstrating whether this is mediated via humoral factors or through cellular activation of circulating cells. What is the biological significance of pyroptosis of Type H endothelial cells as compared to Type L endothelial cells.

To assess mechanisms exogenous to the bone marrow niche, we have analyzed the involvement of the adrenergic signaling post-MI in the bone, which is known to be activated upon MI. This was shown to affect the bone marrow stem cell niche. However, the β 2AR antagonist ICI 118,551 did not prevent infarction induced loss of H type vasculature reduction or expansion of the myeloid population. The results of this experiment are now present on new Supplementary figure 7 and read in the text as follows:

Previous studies showed that MI-associated sympathetic activity and stimulation of the bone niche drives an inflammation-mediated deterioration of the niche^{10,11}. Moreover, increased β 2-adrenergic activity has been shown to substantially influence niche microenvironment by promoting IL-6-dependent CD41+ myeloid progenitor expansion¹⁰. We therefore speculated that blocking beta-2 adrenergic receptor (β 2AR) signalling protects the vascular niche in the bone after MI. We performed MI in the presence of ICI-118,551^{12,13}, a well-known β 2AR antagonist (Supplementary Fig. 8a and Supplementary Fig. 9). However, we could not detect effects on MI-induced reduction of H type endothelium demonstrating that activation of β 2AR signaling does not mediate deterioration of the bone vascular niche (Supplementary Fig. 8b, d) or expansion of the myeloid progenitors (Supplementary Fig. 8c,e). This excludes the contribution of sympathetic humoral signalling to the reduction of H type vasculature after MI.

To assess intrinsic, temporal responses of the bone marrow niche post-MI we have performed scRNA-seq. Here we see strong influences of responses to cytokines (via cytokine signaling associated GO terms) and a notable increase in “positive regulation of cell death” at 28 days post-MI. These approaches provide internal and external vantage points for the response of the bone marrow vascular niche after cardiac insult. Moreover, when the relative expression of genes from GO terms associated with pyroptosis (GO:0070269) vs. apoptosis (GO:0043065) is assessed, there is a clear a higher induction of genes associated with pyroptosis rather than apoptosis when comparing Day7 to Day0 in the *Emcn* expressing cells. Interestingly, this induction of genes associated with the pyroptosis GO term is not strongly upregulated for *Emcn* negative or the unselected general cell population, indicating that *Emcn* expressing cells may be susceptible to pyroptosis after MI.

Comparative expression of constituent genes for GO terms related to pyroptosis and apoptosis.

(a) Average, relative Z-score for expression of constituent genes in pyroptosis comparing *Emcn* expressing cells to all cells. (b) Average, relative Z-score for expression of constituent genes in apoptosis comparing *Emcn* expressing cells to all cells.

Z-score established for each gene among timepoints, normalized to Day0, and aggregated across all genes in the GO term.

To further explore this, we evaluated relative expression of IL-1 β receptor genes and found an increase in the expression of receptor *Il1r1* in *Emcn*^{high} cells. These data further support the potential sensitivity of *Emcn*^{high} cells to IL-1 β signaling. These data appears in Supplementary Fig. 7c and reads in the text as follows:

the expression of the IL-1 β receptor was particularly enriched in *Emcn*^{high} cells (Supplementary Fig. 7c)

5. It is unclear whether the control mice received sham thoracotomies without LAD ligation. If not, then it would be difficult to ascribe the post-ligation day 3 alterations in the bone marrow to the MI alone, since the post-surgical inflammatory state would also contribute. However, the gradually increased effect on the bone marrow by 28-day data argues in favor of the authors' conclusions and mitigates this concern.

The control mice did not receive sham thoracotomies without LAD ligation. To address this issue, we have analyzed the effect of sham thoracotomies on the vascular niche in the bone. Sham animals showed H type vessels length and CD41 positive cell numbers similar to the controls.

Human studies:

1. The authors contend that the scRNA analysis of 1 healthy and 1 ischemic heart failure patient support the relevance of the murine findings to human. Strictly speaking, this is true, but it does not provide robust support, since two adult humans of similar and presumably advanced age may have vastly different clinical backgrounds that give unknown confounders to findings in the blood and bone marrow (i.e. diabetes, smoking, RA, and any inflammatory condition). This could be added to supplementary 1 alongside the HF patient with scRNA data.

We have amended the clinical data table accordingly.

2. Likewise, no clinical data is given for the 8 "healthy" in the FACS analysis other than their median age and that they had no evidence of CAD. This data could also be added to supplementary table 1. The fact that their median age being about 1/2 that of the heart failure patients is also concerning and might not be statistically sufficient to establish that the type H depletion is age-independent in humans as clearly shown in mice. Given the difficulty of obtaining ideal controls through this challenging procedure, the reviewer feels this is an acceptable limitation in the study, but should be acknowledged in the text or tables.

We have amended the clinical data table accordingly to address this limitation.

Minor points:

• The final paragraph regarding the link between heart failure and fractures is supported only by retrospective data in Circulation, which could be confounded by low-activity in HF individuals/ low sunlight. Since the paper does not directly address bone strength in human or mouse data, it might be stronger without this as the final point in the conclusion.

We have removed the last sentence of the discussion.

• 17 out of 18 humans were male. All mice were male. The male/female split of the healthy controls is not given. While CAD/ischemic HF does show is predominantly found in males, this seems appropriate, but please add the gender of the controls as well since estrogen is known to have its own protective effects on endothelium.

We have added the gender information to the Supplementary table that contains the clinical information of the patients.

REVIEWERS' COMMENTS

Reviewer #1 (Remarks to the Author):

All of my comments have been addressed.

Reviewer #2 (Remarks to the Author):

All the concerns of this reviewers were addressed.

Reviewer #3 (Remarks to the Author):

The authors have performed significant additional experiments and have satisfactorily resolved major concerns.